# R*: Efficient Reward Design via Reward Structure Evolution and Parameter Alignment Optimization with Large Language Models

Pengyi Li [1]   Jianye Hao [1]   Hongyao Tang [1]   Yifu Yuan [1]   Jinbin Qiao [1]   Zibin Dong [1]   Yan Zheng [1]

## Abstract

Reward functions are crucial for policy learning. Large Language Models (LLMs), with strong coding capabilities and valuable domain knowledge, provide an automated solution for high-quality reward design. However, code-based reward functions require precise guiding logic and parameter configurations within a vast design space, leading to low optimization efficiency. To address the challenges, we propose an efficient automated reward design framework, called **R***, which decomposes reward design into two parts: reward structure evolution and parameter alignment optimization. To design high-quality reward structures, R* maintains a reward function population and modularizes the functional components. LLMs are employed as the mutation operator, and module-level crossover is proposed to facilitate efficient exploration and exploitation. To design more efficient reward parameters, R* first leverages LLMs to generate multiple critic functions for trajectory comparison and annotation. Based on these critics, a voting mechanism is employed to collect the trajectory segments with high-confidence labels. These labeled segments are then used to refine the reward function parameters through preference learning. Experiments on diverse robotic control tasks demonstrate that R* outperforms strong baselines in both reward design efficiency and quality, surpassing human-designed reward functions.

## 1. Introduction

Deep Reinforcement Learning (DRL) (Sutton & Barto, 1998) has shown remarkable success in various sequential decision-making problems, such as drone racing (Kaufmann et al., 2023a), robotic locomotion (Radosavovic et al., 2024), manipulation (Aguinaco et al., 2023; Yuan et al., 2025), navigation (Zhu & Zhang, 2021), and protein structure design (Lutz et al., 2023). Although these achievements demonstrate the potential of DRL, its learning process often suffers from instability, leading to suboptimal policies or even policy collapse (Arulkumaran et al., 2017). A key factor for this problem is the low quality of reward signals (Hare, 2019; Silver et al., 2021; Eschmann, 2021), which are often sparse and deceptive. Designing high-quality reward signals typically requires extensive domain-specific knowledge, resulting in significant design costs (Sutton & Barto, 1998). Even with carefully hand-crafted reward functions, it remains difficult to guarantee efficient guidance for policy learning (Booth et al., 2023a). How to provide high-quality reward signal guidance remains an open challenge (Menell et al., 2017; Icarte et al., 2022).

To address the challenge of reward design, many methods are proposed from different perspectives, which can be broadly classified into three categories: Inverse RL (Arora & Doshi, 2021), RL from Human Feedback (RLHF) (Kaufmann et al., 2023b), Reward Function Generation (Ma et al., 2024). Inverse RL begins by collecting expert demonstrations and then learns the reward function based on these demonstrations. This approach relies heavily on the coverage and quality of the expert data (Ng & Russell, 2000; Ho & Ermon, 2016). RLHF-related methods incorporate human feedback into RL by comparing trajectories, assigning preference labels, and training a reward model (Lee et al., 2021). This process typically requires the continuous involvement of human experts to provide preference annotations (Lee et al., 2024). The final category focuses on directly constructing reward function codes. The most straightforward approach is to manually design reward functions through trial-and-error (Booth et al., 2023b; Knox et al., 2023). Some early methods employ Evolutionary Algorithms (EAs) (Bäck & Schwefel, 1993; Bäck, 1996; Vikhar, 2016; Zheng et al., 2019) with manually designed operators for reward function design (Niekum et al., 2010; Faust et al., 2019). However, these methods still heavily depend on expert priors.

Recently, Large Language Models (LLMs) have demon-

[1]College of Intelligence and Computing, Tianjin University, China. Correspondence to: Jianye Hao <jianye.hao@tju.edu.cn>.

*Proceedings of the 42nd International Conference on Machine Learning*, Vancouver, Canada. PMLR 267, 2025. Copyright 2025 by the author(s).

strated extensive domain knowledge and remarkable coding capabilities (Singh et al., 2023; Ichter et al., 2022; Wang et al., 2024a;b). Some works use LLMs to directly generate the code of reward function, significantly reducing human labor (Xie et al., 2024; Zeng et al., 2024). However, relying solely on LLM-generated reward functions still faces challenges such as large search spaces and limited robustness. Concurrently, Evolutionary Reinforcement Learning (ERL) (Drugan, 2019; Sigaud, 2022; Bai et al., 2023; Li et al., 2024a)—which combines EAs with RL—has achieved significant improvements in exploration efficiency and learning robustness (Khadka & Tumer, 2018; Zheng et al., 2019; Gupta et al., 2021; Hao et al., 2023; Li et al., 2023; 2024b;c) across various domains. Accordingly, Eureka (Ma et al., 2024) applies the Synergistic Optimization (Li et al., 2024a) idea in ERL for reward generation: EAs evolve a population of LLM-generated reward functions, while RL is employed to train policies for evaluation to select best reward function. Although Eureka demonstrates state-of-the-art performance across a range of tasks, two limitations stand out: 1) **Greedy Exploitation**: Eureka improves solely based on the best reward function at each iteration, which may lead to suboptimal results on reward design. Other potentially valuable candidates are discarded without being fully utilized. 2) **Suboptimal Parameter Assignment**: Reward functions often include many parameters, such as weights or coefficients. Eureka relies on LLMs to set them directly, which is often suboptimal. As a result, the reward functions may fail to provide precise guidance.

To address the above challenges, we propose R\*, a novel LLM-driven evolutionary framework that decouples the reward function design process into structure evolution and parameter optimization. To optimize reward function structures, R\* employs LLMs to generate modular-based reward functions and maintains a reward function population for evolutionary refinement. The reward function fitness is defined by the performance of the policy that the reward function guides, i.e., success rate. R\* leverages the LLM reflection mechanism to improve reward functions through modular modifications, deletions, and additions. To fully exploit high-quality reward functions, R\* maintains a reward function archive and selects parent candidates based on their fitness. By employing the module-level crossover operator, R\* promotes efficient exploration of the reward design space and fully leverages the reward function candidates. To address the challenge of parameter configuration, we propose parameter alignment optimization, which aligns the pairwise trajectory rankings induced by the reward function with ground-truth preferences. This enables more efficient parameter optimization. To achieve this goal, a reliable labeled trajectory dataset is necessary. Some previous works rely on LLMs for numerical comparisons (Zeng et al., 2024), but they often suffer from severe hallucination issues, lead-

ing to low-quality labels. Others utilize VLMs for comparisons (Liu et al., 2024), but existing VLMs struggle to perform reliable analysis of visual inputs. Moreover, quality differences typically exist only in specific segments of a trajectory, and the above methods struggle to effectively extract these fine-grained segments. To solve above problems, we propose the critic-based step-wise voting mechanism. This method uses LLMs to generate critic functions that evaluate trajectories at the state level. The evaluation results are aggregated through population-based voting. Next, trajectory segments are extracted by grouping consecutive steps with consistent labels. Finally, we optimize reward function parameters within the population using pairwise preference loss. Experiments across 8 robotic control tasks demonstrate that the reward functions designed by R\* outperform those generated by other strong baselines in terms of both final performance and convergence efficiency.

We summarize our contributions below:

- We propose a novel LLM-based reward design framework R\* with two key components: reward structure evolution and parameter alignment optimization.

- We design modular-based reward functions and employ LLMs and module-level crossover operations for efficient structure search.

- We propose the critic-based step-wise voting mechanism for data collection and employ pair-wise preference loss for parameter optimization.

- We empirically show that R\* outperforms other strong baselines and surpasses human-designed reward functions in various tasks.

## 2. Background

**Reinforcement Learning** can be formalized as a Markov Decision Process (MDP) (Puterman, 1990) which can be defined by a tuple $\langle \mathcal{S}, \mathcal{A}, \mathcal{P}, \mathcal{R}, \gamma, T, \rho \rangle$, where $\mathcal{S}$ is the state set, $\mathcal{A}$ is the action set, $\mathcal{P} : \mathcal{S} \times \mathcal{A} \times \mathcal{S} \to \mathbb{R}$ is the transition function, $\mathcal{R} : \mathcal{S} \times \mathcal{A} \to \mathbb{R}$ is the reward function, $\gamma \in [0, 1)$ is the discounted factor and $T$ is the horizon. $\rho$ represents the distribution of the initial state. The agent interacts with the environment by performing its policy $\pi : \mathcal{S} \to \mathcal{A}$. RL (Sutton & Barto, 1998) completes the task by maximizing the expected discounted cumulative reward $J(\pi) = \mathbb{E}_{a_t \sim \pi(s_t), s_{t+1} \sim \mathcal{P}(s_{t+1}|s_t, a_t)}[\sum_{t=0}^{T} \gamma^t r_t]$, where $r_t = \mathcal{R}(s_t, a_t)$ and $s_0 \sim \rho$.

In this paper, we aim to design a reward function in code form that guides policy learning to maximize task success rates. To utilize the reward function codes, we need the variable information as function inputs. To simplify notation, we assume that these variables are included in $s$.

**Preference Learning**. In our work, we employ preference learning for the optimization of reward function parameters. We define a segment $\sigma$ as a sequence of step-indexed states $\{\mathbf{s}_k, ..., \mathbf{s}_{k+H-1}\}$ with length $H$. Given a pair of segments $(\sigma^0, \sigma^1)$, we provide a label indicating which segment is preferred, so the preference $y_{\text{comp}}$ could be indicated $\sigma^0 \succ \sigma^1, \sigma^1 \succ \sigma^0$. We discard data with equal preferences, and the remaining labels are encoded as binary vectors, i.e., $y_{\text{comp}} \in \{(1,0), (0,1)\}$, and stored as tuples $(\sigma^0, \sigma^1, y_{\text{comp}})$ in the trajectory buffer $D_T$. Subsequently, we use the annotated data to optimize the reward function $F_\psi$, where $\psi$ represents the parameters extracted from $F$, such as the parameters within the reward module and the weights between modules.

Following the Bradley-Terry model (Bradley & Terry, 1952), we model the preference based on $F_\psi$ as follows:

$$P_\psi[\sigma^1 \succ \sigma^0] = \frac{\exp(\sum_t F_\psi(\mathbf{s}_t^1))}{\sum_{i \in \{0,1\}} \exp(\sum_t F_\psi(\mathbf{s}_t^i))}, \quad (1)$$

where $\sigma^i \succ \sigma^j$ denotes the event that segment $i$ is preferable to segment $j$ and $(\mathbf{s}_t^i, \mathbf{a}_t^i) \in \sigma^i$. To align the reward function with the preferences, The update of $F_\psi$ is transformed into minimizing the following cross-entropy loss:

$$\mathcal{L}(\psi, D_T) = - \mathbb{E}_{(\sigma^0, \sigma^1, y_{\text{comp}}) \sim D_T} \Big[ y(0) \log P \left[ \sigma^0 \succ \sigma^1 \right]$$
$$+ y(1) \log P \left[ \sigma^1 \succ \sigma^0 \right] \Big],$$
$$(2)$$

where the terms $y(0)$ and $y(1)$ refer to the values at index 0 and 1 of the label vector $y_{\text{comp}}$, respectively.

## 3. Related Work

Various methods are proposed from different perspectives to construct high-quality reward signals. IRL learns the reward function from the provided expert demonstrations (Adams et al., 2022), including methods such as max-margin methods (Ng & Russell, 2000), Bayesian methods (Ramachandran & Amir, 2007), and maximum entropy methods (Ziebart et al., 2008). In addition, GAIL (Ho & Ermon, 2016) uses GAN (Goodfellow et al., 2014) to approximate reward function for policy learning. However, these methods are often sensitive to the quality and distribution of the data.

RL from Human Feedback (RLHF) (Kaufmann et al., 2023b; Dong et al., 2023; Yuan et al., 2024) leverages human feedback to guide the learning process of RL agents. PB-RL (Christiano et al., 2017) collects preference data through trajectory pair comparisons with human experts, which is then used to learn a reward model that guides RL learning. PEBBLE (Lee et al., 2021) combines unsupervised learning to pretrain the agent and integrates off-policy RL. Additionally, some works focus on improving the feedback mecha-

nism to improve both efficiency and accuracy (Sharma et al., 2022; Guan et al., 2023; Zhou et al., 2025). Besides, some methods manually design reward functions through trial and error (Booth et al., 2023b; Knox et al., 2023), while others optimize reward functions using EAs with predefined operators and templates (Faust et al., 2019; Niekum et al., 2010). These methods typically depend on domain knowledge from human experts.

With Large Language Models (LLMs) demonstrating strong domain knowledge and coding abilities, several works try to generate reward function code through LLMs. For example, Text2Reward (Xie et al., 2024) generates reward functions by calling pre-defined interfaces through LLMs based on task descriptions. Eureka (Ma et al., 2024) constructs a reward function population with LLMs and proposes an evolutionary framework for iterative improvement. While these works achieve impressive results, the vast design space leads to inefficient reward function optimization and limited learning robustness. In addition, there are some works on reward shaping (Ladosz et al., 2022), such as curiosity-driven mechanisms (Burda et al., 2019), which typically focus on designing intrinsic rewards to enhance exploration. In contrast, our work focuses on generating reward functions from scratch. Our work, like Eureka, also falls within the Synergistic Optimization (Li et al., 2024a) branch in ERL: EAs are used to iteratively optimize the reward functions, while RL handles policy learning for population evaluation.

## 4. Method

This section presents our framework, R*, for efficient reward function design. The key idea of R* is to decompose the reward design into two parts: reward structure evolution and parameter alignment optimization. The former is responsible for searching the functional components that should be included, while the latter focuses on efficiently optimizing the parameters within and across modules.

### 4.1. Framework Overview

We first present the overall framework of R* to gain a holistic understanding. The optimization flow is shown in Figure 1, which consists of two stages: the initialization stage and the evolutionary improvement stage.

The initialization stage consists of two key steps: (1) Initializing the reward function population, which provides the foundation for subsequent reward function design; (2) Initializing the critic population, which is used to collect trajectory segment pairs with high-quality labels.

1. **Reward Population Initialization**. R* leverages LLM to generate the initial reward function population $\mathbb{P}_{\text{Reward}}$ by providing the task description $L$ and

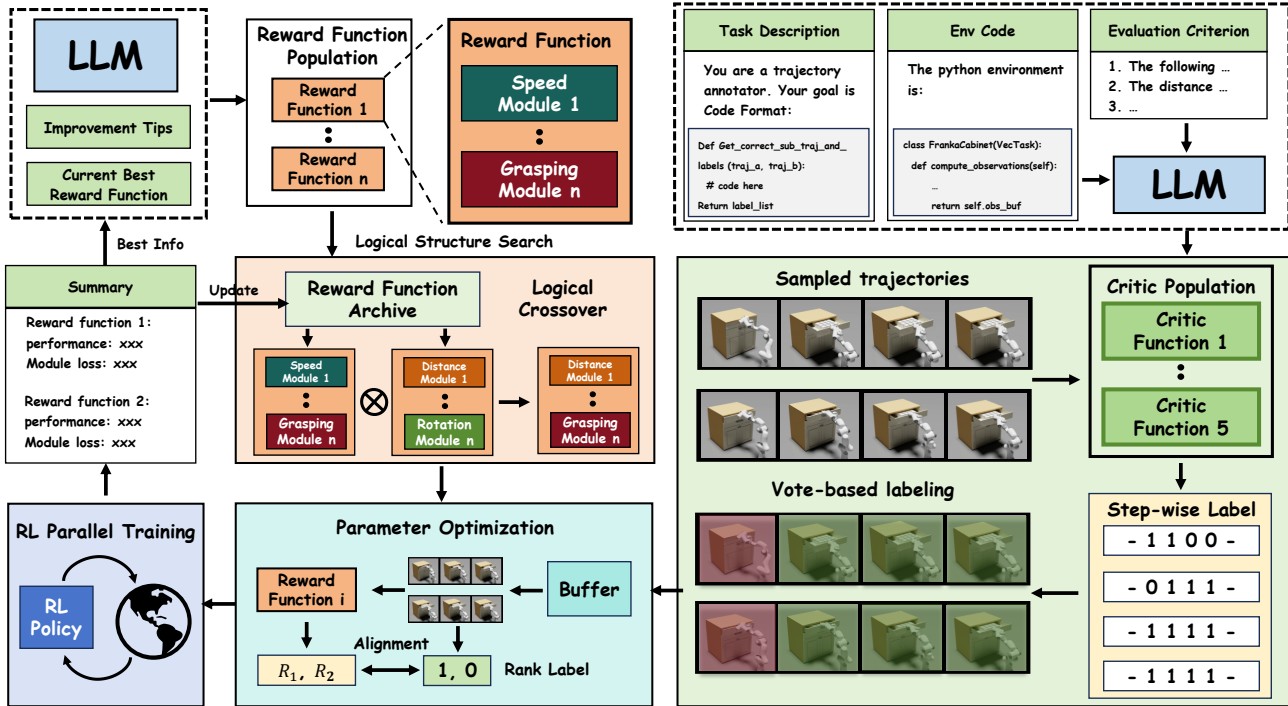

*Figure 1.* The optimization flow of R\*. The left side depicts the main reward optimization process, while the right side illustrates trajectory annotation for parameter optimization. (Left) First, generate reward population using LLM. Then, select superior parents from the reward function archive for module-level crossover. Next, perform the parameter optimization. Following this, train RL parallelly. After training, update the reward function archive, and the training results are summarized to select the best individual for LLM reflection. (Right) Generate the critic function population using LLM. The individuals perform step-level comparisons on the sampled trajectories. A voting mechanism is then used to annotate and extract trajectory segments with high-confidence labels for parameter optimization.

environment code information $C$. To facilitate more efficient structure search in subsequent stages, we design the prompt $P_{\text{reward}}$ to guide the LLM in generating reward functions composed of multiple reward modules.

2. **Critic Population Initialization**. In addition to the reward population, R\* also needs to generate a critic population $\mathbb{P}_{\text{Critic}}$, with each individual represented as a code function. These critics are used to compare trajectories and provide step-wise comparison labels. Similar to the reward population, R\* takes task descriptions $P$, environment code $C$, and the prompt $P_{\text{critic}}$ as inputs to generate the critic population $\mathbb{P}_{\text{Critic}}$. A detailed introduction will be provided in subsection 4.3.

After completing the initialization stage, we proceed to the reward function evolutionary improvement stage, which consists of three steps: population evaluation, evolution, and parameter optimization.

1. **Reward Population Evaluation**. For each reward function $F_i$ in the reward population $\mathbb{P}_{\text{reward}}$, we train PPO (Schulman et al., 2017) in parallel to solve the task

using Isaac Gym (Makoviychuk et al., 2021). During training, we record the success rates of the learned policies, along with other relevant statistics, such as the mean, maximum, and minimum values of each reward module. The reward functions, along with their fitness values, are then stored in the reward function archive $D_F$. The success rate of the policy serves as the reward function fitness.

2. **Reward Population Evolution**. We select the reward function $F_{\text{best}}$ with the highest fitness, along with its associated learning information, to serve as feedback. Through the LLM reflection mechanism, we perform evolution operations such as modifying, deleting, or adding modules to improve the reward function, thereby creating a new improved reward function. To fully exploit the high-quality reward functions discovered, we select parent reward functions from the reward function archive $D_F$ based on their fitness. Module-level crossover is then applied to these parents to generate new individuals, facilitating further exploration of the reward design space.

3. **Reward Parameter Optimization**. Using the critic

population $\mathbb{P}_{\text{Critic}}$, we extract trajectory segments and apply comparison labels to the segments, creating a labeled trajectory buffer $D_T$. The reward functions are then optimized based on $D_T$, allowing them to better evaluate segment quality and guide policy learning.

The initialization is performed at the beginning of the algorithm, followed by an iterative process of reward function evolutionary improvement. In the following subsections, we provide a detailed introduction of the two key components.

## 4.2. Reward Structure Evolution

Reward functions are typically composed of multiple reward modules to guide policy learning. For example, in robot grasping tasks, the reward function should include the reward for guiding the robot arm closer to the target object, the reward for moving the target object, and reward for the arm's stability. These modules serve different objectives. How to design and combine these modules to create high-quality reward functions remains a challenging problem.

Inspired by the previous work (Ma et al., 2024), we first propose the modular-based reward function, where each reward function $F$ is composed of multiple modular components $\{M_1, \cdots, M_m\}$. The design of high-quality reward functions depends both on the internal structure of modules and how these modules are integrated. To achieve efficient optimization, we leverage LLM to improve the reward functions with following three main operators: 1) **In-module Improvement**. Adjusting the reward calculation within modules, such as applying a scaling transformation to distance-guided rewards. 2) **Removing Module**. Based on feedback from each module during the learning process, removing ineffective reward modules from the reward function. 3) **Adding Module**. Introducing new reward modules to improve the reward function.

The above process relies on LLMs but does not fully leverage the high-quality reward functions discovered. To address this, we design a selection operator and a module-level crossover operator. Specifically, we first maintain a reward function archive $D_F$ to store the high-quality reward functions discovered. We select two parents based on the fitness and then perform a crossover operation by inserting a module from one parent into the other, forming a new reward function. We formulate the operation below:

$$F_{\text{new}} = \{M_{i,1}, \cdots, M_{i,m}, M_{j,m}\} = \text{Crossover}(F_i, F_j). \tag{3}$$

By leveraging the inherent randomness of the crossover operation, we can fully leverage high-quality reward functions to facilitate a more comprehensive exploration of the design space. The above process can be enhanced using LLMs in two ways: by generating reward functions to replace random module swaps, and by filtering out unreasonable ones—thus

reducing unnecessary exploration. This paper primarily provides an initial exploration of the method. Given that LLMs require additional token overhead, we leave this aspect for future work.

## 4.3. Parameter Alignment Optimization

In the previous subsection, we generate the initial reward population $\mathbb{P}_{\text{Reward}}$. However, configuring parameters within these reward functions remains challenging. Since the function parameters are provided by LLMs, this black-box configuration approach makes it difficult to guarantee accurate and efficient guidance, leading to issues in both in-module parameter settings and inter-module weight coordination. To address this challenge, we propose ranking-based parameter alignment optimization, which consists of two key steps: automated trajectory annotation and parameter alignment optimization. Automated trajectory annotation is used to collect trajectory segments and assign comparison labels, while parameter alignment optimization optimizes reward parameters with the labeled data. Below, we first describe how to extract trajectory segments and assign high-confidence labels.

To collect data automatically, previous methods typically rely on LLMs or VLMs to perform comparisons based on numerical states or images. However, these methods face three key problems: 1) Numerical comparisons rely on manually specified state features, and the instability of LLMs often leads to unreliable outcomes; 2) VLMs frequently provide incorrect labels due to their limited capabilities and the inherent complexity of the task; 3) It is difficult to directly assess the relative quality of two trajectories, as one may outperform the other in some segments while underperforming in others. Thus labeling the entire trajectory as a whole can result in low-quality annotations.

To address the above challenges, we propose the critic-based comparison voting mechanism, which constructs a population of rule-based critic functions for step-wise comparison. Specifically, we input the task description $L$, the environment information $C$, and the prompt $P_{\text{critic}}$ into the LLM, allowing it to automatically generate the critic functions. The input of the critic functions consists of two trajectories to be compared, which contain information such as variable names and their corresponding values, e.g., "block_right_handle_pos=(0.0, 0.5, 0.0)". Using above information, the LLM can quickly understand the input and construct a critic function. Regarding the function details, we require it to compare the corresponding-step information of the two trajectories and assign labels of 1, 0, or -1. We require that the function assigns a label of 1 or -1 only when all metrics of the current step of trajectory A are either better or worse than those of trajectory B; otherwise, it assigns a label of 0. To further enhance reliability, we generate a

population of critics rather than relying on a single critic function. The final label is assigned only when at least half of the critics reach agreement. After obtaining the final step-wise labels, we group consecutive steps with the same label (i.e., 1 or -1) into segments. Segments with a length greater than 5 are added to the buffer $D_T$ for subsequent optimization.

After obtaining the labeled segments, we conduct parameter alignment optimization for the reward functions within $\mathbb{P}_{\text{Reward}}$. The data is then split into 70% for training and 30% for validation, and we optimize the function parameters with preference loss in Equation 2. Through parameter optimization, each reward function can accurately evaluate the trajectory quality, facilitating efficient policy guidance.

### 4.4. R* Pseudocode

To provide a clear overview of R*, we present the pseudocode in Algorithm 1. Specifically, we begin with the *Initialization Stage*, where the task description $L$, environment code $C$, and prompts $P_{\text{critic}}$ and $P_{\text{reward}}$ are provided as contextual inputs to the LLM, which then generates two populations: $\mathbb{P}_{\text{Critic}}$ and $\mathbb{P}_{\text{Reward}}$ (line 3). Next, we proceed to the *Evolutionary Improvement Stage*. We first sample parents $F_{p_1}$ and $F_{p_2}$ from $D_T$ and apply the crossover operator to generate multiple offspring, which are then added to the reward function population $\mathbb{P}_{\text{Reward}}$ (line 6-8). Subsequently, we sample new trajectories and annotate them using $\mathbb{P}_{\text{Critic}}$, incorporating the labeled segments into $D_T$. Based on the collected data, we optimize the reward parameters according to Equation 2 (line 10-12). We then conduct population evaluation, update the reward archive $D_F$ and obtain the best reward function $F_{\text{best}}$ (line 14-15). Finally, we employ the LLM to refine the reward functions within the population (line 17). The above shows the complete optimization process. In the next section, we conduct a comprehensive experimental evaluation of R*.

## 5. Experiments

### 5.1. Environment Setting

We compare R* and the baseline methods on diverse robotic manipulation tasks to evaluate their reward generation capabilities. All methods used GPT-4o as the backbone LLM for generation. The test tasks are derived from the Isaac Gym (Makoviychuk et al., 2021) and the Bidextrous Manipulation (Dexterity) benchmark (Chen et al., 2022). These tasks cover manipulation tasks using robotic arms and single dexterous hands, as well as tasks requiring dual dexterous hands to perform complex operations, ranging from object handover to rotating a cup by 180 degrees. Our implementation is primarily based on Eureka (Ma et al., 2024), the population size is set to 16, and the number of evolution

---

**Algorithm 1** R* Framework

1: **Require**: Task description $L$, environment code $C$, reward function prompt $P_{\text{reward}}$, critic function prompt $P_{\text{critic}}$, coding LLM $\mathbb{LLM}$, reward function archive $D_F$, trajectory buffer $D_T$

2: **Hyperparameters**: Evolution iteration $K$, crossover individual number $n_c$, reward population size $n$, critic population size $c$.
   Stage I: The Initialization Stage

3: Initialize reward and critic populations:
   $\mathbb{P}_{\text{Reward}} = \mathbb{LLM}(L, C, P_{\text{reward}})$
   $\mathbb{P}_{\text{Critic}} = \mathbb{LLM}(L, C, P_{\text{critic}})$
   Stage II: The Evolutionary Improvement Stage

4: **for** each iteration **do**

5:    # Modular Crossover

6:    Sample parents $F_{p_1}, F_{p_2} \sim D_F$

7:    Construct new functions through crossover
      $F' = \text{Crossover}(F_{p_1}, F_{p_2})$

8:    Add new functions into $\mathbb{P}_{\text{Reward}}$

9:    # Parameter Alignment Optimization

10:   Sample trajectories and label them using $\mathbb{P}_{\text{Critic}}$.

11:   Add labeled data into $D_T$.

12:   Optimize parameters based on $D_T$ using Eq. 2

13:   # Reward Population Evaluation

14:   Evaluate with parallel PPO training

15:   Summarize training result, update the reward function archive $D_F$ and get the best reward function $F_{\text{best}}$

16:   # Reward Population Evolution with LLM

17:   Improve reward population with LLM reflection
      $\mathbb{P}_{\text{Reward}} = \text{Reflection}(L, C, P_{\text{reward}}, F_{\text{best}})$

18: **end for**

---

iterations is set to 5. The key difference lies in how individuals in the population are generated. Specifically, among the 16 individuals, 12 are generated or improved by the LLM, and 4 are generated through the module-level crossover. To guarantee that each experiment fits within a 40 GB GPU memory budget (e.g., using two NVIDIA 3090 or 4090 GPUs), we adjust the number of parallel environment instances for certain tasks, which significantly increases the task learning difficulty. Detailed configurations are listed in Appendix A. To ensure a fair comparison, all other hyperparameter settings and related prompts remain consistent with Eureka. RL training is conducted using PPO under identical configurations, as detailed in Appendix A. All statistics are obtained from 5 independent runs. We report the average with 95% confidence interval.

### 5.2. Baselines

**Eureka** (Ma et al., 2024) is a state-of-the-art LLM-based method for reward design. It constructs reward functions

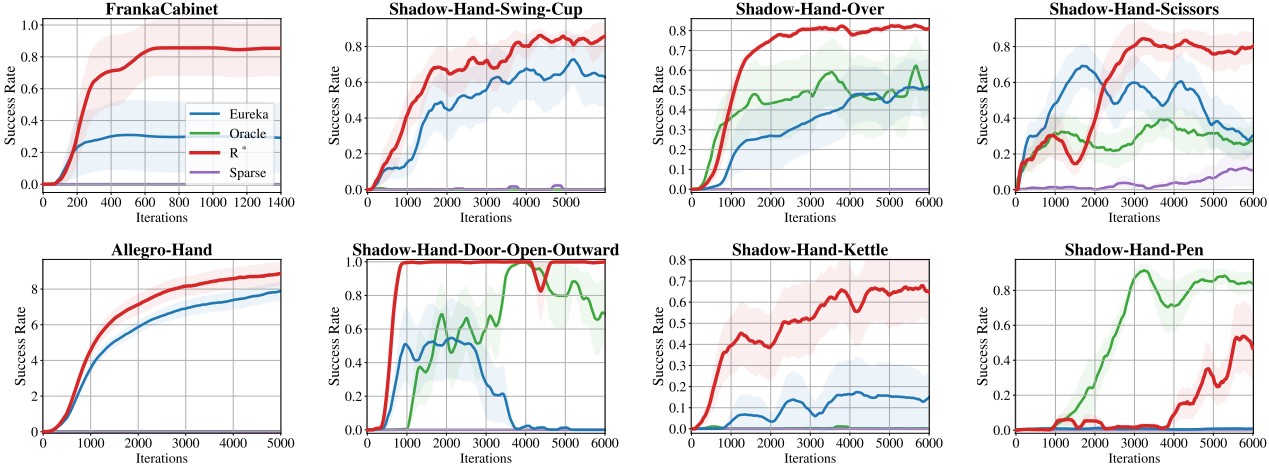

*Figure 2.* Performance comparison on various manipulation tasks. R* significantly outperforms other baselines and surpasses human-designed rewards in most tasks.

from scratch without incorporating any human prior knowledge. Similar to our method, Eureka maintains a reward population constructed by an LLM and improves the reward functions within the population through a reflection mechanism.

**Oracle**: In these tasks, reward functions manually crafted by human experts are used to guide policy learning. These reward functions represent expert-level human reward engineering. We use these manually designed reward functions for policy training as baselines for comparison.

**Sparse**: Based on the success criteria of the tasks, a sparse reward is provided, where a reward of 1 is given only when the task is successfully completed, and 0 is returned at all other times.

### 5.3. Performance Evaluation

We first evaluate R* and baselines on the diverse robotic manipulation tasks with different characteristics. The experimental results are shown in Figure 2. We observe that the reward functions generated by R* significantly outperform the strong baseline Eureka in guiding policy learning. Furthermore, R* exhibits more stable learning performance, whereas Eureka suffers from policy collapse in some tasks. In addition, R* achieves superior results compared to the oracle reward function in most tasks, indicating that the reward functions designed by R* can match or even surpass those designed by human experts. Sparse rewards generally fail to effectively guide policy learning and are unable to yield effective policies in almost all tasks, further highlighting the importance of high-quality reward guidance. Furthermore, we perform experiments under the original parallel environment settings used in Eureka, where R* also demonstrates a significant performance advantage. Detailed

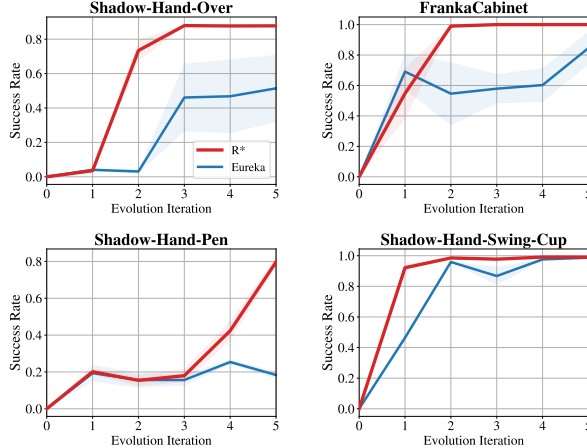

*Figure 3.* Comparison of the best policy success rate per generation between R* and Eureka. R* achieves higher evolutionary efficiency.

results are presented in Appendix A.

Since both R* and Eureka are evolutionary-based reward function generation methods with identical settings, we compare the best-performing individuals with the highest success rates in the population at each generation, as shown in Figure 3. The experimental results demonstrate that R* achieves higher success rates at earlier generations, indicating a more efficient evolutionary efficiency. This further validates the efficiency of R*.

### 5.4. Analysis and Ablation Study

In this subsection, we further analyze R* to answer the following questions:

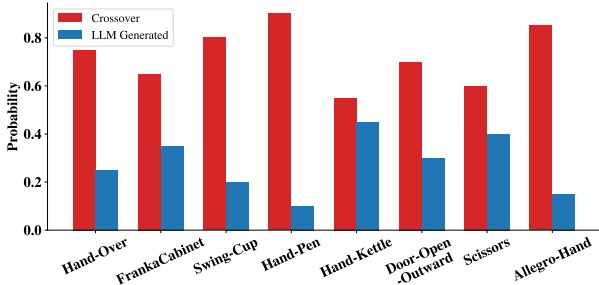

Figure 4. Performance comparison on various manipulation tasks. Individuals generated by crossover are more likely to achieve better performance.

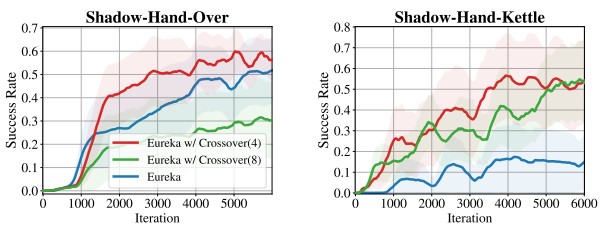

Figure 5. Performance comparison across different numbers of crossover individuals. A high proportion of crossover individuals can lead to performance degradation.

**Q1.** Does the module-level crossover in R* significantly enhances algorithm performance? How does its impact compare to that of the LLM in reward function design, and which component plays a more critical role?

**Q2.** Can the critic population constructed by R* provide high-quality data annotations? Additionally, does the parameter optimization in R* further improve the quality of the reward function?

To answer Q1, we compute the ratio of generations in which the best policy originates from crossover, and compare it to the ratio of generations in which the best policy originates from LLM-driven refinement. The experimental results in Figure 4 show that the probability of best policies originating from crossover exceeds 50% in most tasks, with some tasks surpassing 80%. Notably, only 4 individuals are generated through crossover, whereas 12 individuals are refined and generated by the LLM. This demonstrates that leveraging prior superior reward functions for further optimization facilitates a more efficient search for high-quality reward functions. Additionally, we conduct an ablation study on crossover. As shown in Figure 6, removing crossover leads to a decline in the performance of R*, further demonstrating the effectiveness of module-level crossover.

Based on the above experiments, we pose a new question: Does increasing the number of crossover-generated indi-

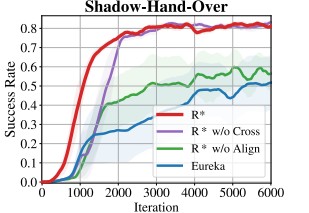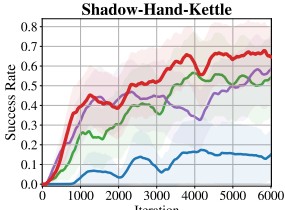

Figure 6. Ablation study on the crossover operator and alignment optimization.

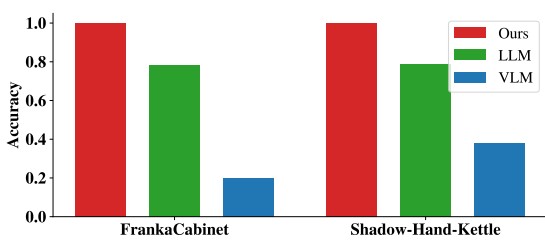

Figure 7. Accuracy achieved by different labeling methods. Our method achieves the best accuracy in data annotation, while VLM has the lowest accuracy.

viduals further enhance performance? To explore this, we increase the number from 4 to 8. The experimental results in Figure 5 show that rather than improving performance, this adjustment leads to a decline, sometimes even underperforming the baseline. The primary reason is that crossover primarily integrates and refines existing high-quality reward functions, relying on the LLM-generated reward functions as a foundation. Reducing the number of individuals generated by the LLM significantly weakens the algorithm's exploration capability, hindering its ability to discover high-quality reward functions and ultimately resulting in degraded performance.

To answer Q2, we first compare several different data annotation methods, including: 1) Using images as inputs to GPT-4o (VLM); 2) Providing key information as context to GPT-4o (LLM); 3) Utilizing our proposed critic population (Ours). We begin by collecting 100 paired comparison samples for both the FrankaCabinet and Shadow-hand-kettle tasks, which are manually labeled by human annotators based on numerical comparisons. The accuracy results in Figure 7 show that our method achieves 100% accuracy, whereas the LLM-based approach performs worse, particularly when the numerical differences are small. Moreover, the VLM usually fails to make correct distinctions and outputs 0. These results demonstrate that the critic-based voting mechanism provides more reliable data annotation.

Furthermore, as shown in Figure 6, removing parameter alignment optimization leads to a significant performance decline in R*. This demonstrates that directly using LLM-

generated parameters may result in suboptimal policies, further demonstrating the effectiveness of the proposed parameter alignment optimization.

We also provide the generated reward functions and critic functions for several tasks. Due to space limitations, please refer to Appendix B for details.

## 6. Conclusion

We propose an efficient framework R* for automatic reward function generation, which consists of two key components: Reward Structure Evolution and Parameter Alignment Optimization. To design high-quality reward function structures, R* leverages LLMs to generate a population of modular-based reward functions. The reward function fitness is defined as the policy performance it guides. We leverage the LLM's reflection mechanism and the module-level crossover for efficient exploration and exploitation of reward functions. To address the parameter configuration challenge, we introduce a critic-based voting mechanism for step-level labeling to collect trajectory segments. We then optimize the reward function population using preference loss based on the collected data. Experiments on eight diverse manipulation tasks demonstrate that R* significantly outperforms other strong baselines and surpasses human-expert-designed reward functions.

## 7. Limitations & Future Work

In this paper, the experiments primarily focus on simulation-based manipulation and dexterous hand control tasks, where low-level information can be directly accessed. For real robot control, a real2sim2real paradigm can be adopted. Since real-world information from the robot and its environment are available, training can proceed by aligning the information from the real robot with that in simulation. This enables direct deployment of the trained policies in real-world scenarios. For tasks where low-level information is inaccessible—such as non-invasive control tasks——only visual observations (i.e., images) are typically available. We believe that the key to applying R* in such cases lies in effective information extraction. In these scenarios, a detection model (e.g., YOLO) can be used to identify and extract relevant features from visual input. Once these key features are obtained, the subsequent reward generation and policy training processes are consistent with those used in tasks where low-level information is accessible.

Existing methods, including Eureka, Text2Reward, and our proposed R*, do not support image-based inputs and are applicable only when all low-level information is fully accessible. This constitutes a key limitation of current approaches. We believe that incorporating image-based inputs can improve generalizability and enable application to a wider range of real-world scenarios, making it a promising direction for future research.

## Impact Statement

This paper presents work whose goal is to advance the field of Machine Learning. There are many potential societal consequences of our work, none which we feel must be specifically highlighted here.

## Acknowledgments

This work is supported by the National Natural Science Foundation of China (Grant Nos. 62422605, 92370132). We would like to thank all the anonymous reviewers for their valuable comments and constructive suggestions, which have greatly improved the quality of this paper.

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

## A. Implementation Details

All prompt designs in R\* for reward function design are based on the prior work Eureka (Ma et al., 2024), without any additional modifications. The specific design details can be found in the original Eureka paper and the corresponding code repository [1]. The reward function population has a size of 16, while the critic population has a size of 5. The number of crossover individuals is 4 for all tasks. If the reward function generation by the LLM encounters runtime failures (e.g., due to a compilation error), the failed individuals are supplemented with those generated via crossover. For instance, if 2 out of 12 reward functions generated by the LLM fail to compile, the module-level crossover will be used to generate 2 additional functions as a supplement. No extra API calls are required in this process.

Since there is no archive during the first iteration, crossover cannot be applied in the first iteration. Thus we adopt a two-step iteration approach: first evaluate the individuals generated by the LLM, then generate individuals via crossover for evaluation, and finally aggregate all evaluation results to select the best individual. This method does not increase the computational workload, but it does require more time. For the modularization of code, we use the Abstract Syntax Tree (AST) to decompose the reward functions based on the returned reward dictionary. This process returns the code blocks corresponding to each reward component, which are then used for subsequent module-level crossover.

In Eureka, some tasks use a large number of parallel environments, requiring at least 4 RTX 4090 GPUs to run. We reduce the number of parallel environments to ensure that the program can run with only 40GB of GPU memory. However, this reduction significantly increases the learning difficulty for the algorithm. The specific configuration of the number of parallel environments is shown in Table 1. All experiments are conducted under the same configuration. Moreover, we conduct

*Table 1.* Number of parallel environment instances per task

|  | Franka–Cabinet | Swing–Cup | Hand–Over | Hand–Scissors | Allegro–Hand | Door–Open–Outward | Kettle | Pen |
|---|---|---|---|---|---|---|---|---|
| Env number | 4096 | 256 | 512 | 128 | 1024 | 2048 | 128 | 256 |

additional experiments using the original setting for the number of parallel environments. The corresponding results are shown in Table 2. We observe that R\* also outperforms Eureka under the original setting.

*Table 2.* Success rates of Eureka and R\* across different tasks under the original setting.

|  | Franka | Swing-Cup | Hand-Over | Kettle | Scissor | Door-Open-Outward |
|---|---|---|---|---|---|---|
| Eureka | 33% | 53% | 83% | 70% | 100% | 98% |
| R\* | 73% | 96% | 93% | 95% | 100% | 100% |

Here, we provide a detailed explanation of several key aspects. First, regarding the population voting mechanism, each critic function evaluates and ranks the quality of different trajectories. The final output is then determined based on the aggregated results of the five critics. To obtain the final labels, we gradually relax the voting criteria from strict to more lenient. Specifically, at the beginning, a state pair is labeled only if all five critics reach a consensus. We collect 20 trajectory segments, each with a length of at least 5. If the required 20 segments are not collected, the criteria are relaxed—labeling occurs if 4 out of 5 critics agree, and if necessary, further relaxed to 3 out of 5 critics until 20 labeled samples are obtained. Based on the collected data, we optimize parameters according to Equation 2. For the crossover operator, we use a softmax function based on the success rates to select parent individuals. A reward module from one parent is incorporated into another function to generate a new individual. This process is repeated until the required number of individuals is generated.

For parameter optimization, we split the collected data into two parts: 70% as the training set and 30% as the validation set. We perform 1000 iterations of optimization on all training data, evaluating accuracy on the validation set after each training iteration. Finally, we select the parameters that achieve the highest accuracy on the validation set as the final optimized parameters.

To construct the critic function, we use the task description, available variables, and success criteria as inputs to guide the LLM in generating the reward function. After the reward functions are generated, we conduct an additional round of reflective optimization. Details of the prompt design are provided later.

---

[1]https://github.com/eureka-research/Eureka

**Prompt 1: Critic Function Task Description**

You are a trajectory annotator, tasked with distinguishing the quality of two robot control trajectories.

Your goal is to write a data extraction function that extracts fully correct sub-trajectories from two given trajectories and assigns quality labels to them.

The data extraction function can be defined as follows:

{Critic-Function-format}

Make sure any new tensor or variable you introduce is on the same device as the input tensors.

Here is the list of keys for dict: {keys}

!!! Please ensure that all the keys used exist in the list mentioned above.

The output of the function should consist of label list: label list contains the quality labels for the states at the corresponding indices in the trajectory. If the state in a is better than the state in b at a given index, the label is 1; if the state in b is better than the state in a, the label is -1; otherwise, the label is 0.

The code output should be formatted as a python code string: ""'python ... "'".

Some helpful tips for writing the function code:

(1) Comparison Criteria for States at the Same Index: The evaluation criterion for comparing the states at the same index is as follows: state a can only be considered better than state b if it is superior to b in all metrics. For example, if two robotic hands are performing a task, and a's left hand is better than b's left hand, but b's right hand is better than a's right hand, the label should be 0.

(2) Task Decomposition into Phases for More Accurate Evaluation: The task needs to be divided into multiple phases to allow for more accurate evaluation. For example, in a dual-hand operation task, if neither hand has grasped the target yet, then comparing the distances to the target is more appropriate. However, if both hands have grasped the target, then the comparison should focus on the distance to the target itself.

(3) Do not set thresholds; directly perform numerical comparisons on the information, for example, if a < b for all distance, a better.

(4) The provided information consists of raw, unprocessed data. Distance information and other metrics need to be calculated.

(5) Typically, you need to consider two distances: the distance between the manipulator or robotic arm and the block to be manipulated, and the distance between the manipulated block (cube) and the target. Or other distances, such as the distance the drawer is pulled open, the distance between two objects, and so on.

**Prompt 2: Critic Function Design Tips**

The Python environment is {task_obs_code_string}. Write a comparison function for the following task: {task_description}.

The conditions for determining task success are as follows: {task_goal}

First, the task needs to be analyzed, and based on the task, the objects that need to be manipulated should be identified. Then, the variables required to complete the task should be inferred. For example, we need to determine the specific positions where the left and right hands should approach. Simply stating that the hands should approach the object is not efficient.

Please provide an analysis before starting, clearly defining the metrics to be calculated and whether there are more appropriate metrics that could replace them. Finally, check if the code achieves the intended purpose.

The following three types of distances need to be considered:

(1). The (five) distances (not sum) between the five fingers (if existing) of the left hand and the target (left) block or cube.

(2). The (five) distances (not sum) between the five fingers (if existing) of the right hand and the (another right) target block or cube.

(3). Based on the task success criteria, the (rotational) distances between the objects (object and the target positions), as required.

The three distances mentioned above should be treated as three separate metrics and should not be combined pairwise into a single metric.

1. Please ensure that your final success distance metric is correctly represented, and avoid fabricating non-existent target information.

2. In addition, use traj_a[key][:,3] to get values instead of traj_a[key][3], use the numerical distance metrics to compare between trajectories instead of use bool value, as directly judging success metric (success_a = dist_a < 0.2) at the start of the trajectory will always return False, resulting in invalid comparison.

3. Typically, it is unlikely for the left hand to operate on the right side or the right hand to operate on the left side.

4. If the key does not exist in the list, do not create the variable.

**Prompt 3: Critic Function Reflection Tips**

Reflect on the correctness of the above codes: Does the code include the following components, and does the code adhere to the given recommendations?

1. verify whether any components are missing, do not sum any components.

(1). The (five) distances (not sum) between the left five fingers (if existing) of the left hand and the (left) target block or cube.

(2). The (five) distances (not sum) between the right five fingers (if existing) of the right hand and the (another right) target block or cube.

(3). Based on the task success criteria, the (rotational) distances between the objects (object and the target positions), as required.

2. Verify each criterion use the key in the list and follow the code format. Key list: {keys}

3. Use the numerical distance metrics to compare between trajectories instead of use bool value, as directly judging success metric (success_a = dist_a < 0.2) at the start of the trajectory will always return False, resulting in invalid comparison.

4. Reasonableness check: it is unlikely for the left hand to operate on the right side or the right hand to operate on the left side.

Check the code against the four aspects mentioned above one by one and analyze whether they are satisfied. If there are no issues, please output the original code; if there are issues, please output the modified code following the same format {Critic-Function-format}

**Critic Function Format**

```python
def get_correct_sub_traj_and_labels(trajectory_a, trajectory_b):

    """
    Both trajectory_a and trajectory_b are lists, each containing multiple states,
        and each state is represented as a dictionary.
    The label_list is a list of indices that corresponds to the states in the sub-
        trajectory where A is definitively better or worse than B. The label
        indicates the quality of the sub-trajectory: if A is better, the label is "
        Former"; if B is better, the label is "Latter".
    If no valid sub-trajectory can be found between the two trajectories, label_list
        will be an empty list ([])
    """
    traj_a_length = len(trajectory_a)
    traj_b_length = len(trajectory_b)
    assert traj_a_length == traj_b_length

    label_list = []

    for state_index in range(traj_a_length):
        state_a_info = trajectory_a[state_index]
        if metric_1_a >= metric_1_b and metric_2_a  >= metric_2_b and metric_2_a >=
            metric_2_b and ... :
            label_list.append(1)
        elif metric_1_a <= metric_1_b and metric_2_a <= metric_2_b and metric_2_a <=
             metric_2_b and ... :
            label_list.append(-1)
        else:
            label_list.append(0)
    return label_list
```

## B. Generated Function Examples

### B.1. Reward Function Examples

We provide the reward function generated by R\* below. It can be observed that the rewards given by R\* are reasonable, with optimized parameters, unlike other methods that only provide integer or single-decimal values.

### B.2. Critic Function Examples

We provide the LLM-generated critic reward function for Franka-Cabinet and Shadow-Hand-Kettle as follows. Upon observation, we find that while the LLM's overall judgment criteria are generally correct. However, this does not necessarily lead to better performance. For example, proximity alone does not imply that one state is superior to another. We consider further exploring this issue in future work.

**The Reward Function For Franka-Cabinet**

```python
 from typing import Tuple, Dict
import math
import torch
from torch import Tensor
@torch.jit.script
def compute_reward(drawer_grasp_pos: torch.Tensor, franka_grasp_pos: torch.
    Tensor, cabinet_dof_pos: torch.Tensor, cabinet_dof_vel: torch.Tensor) ->Tuple[
        torch.Tensor, Dict[str, torch.Tensor]]:

    # Reward weight hyperparameters
    pos_distance_weight = 1.5603480339050293
    drawer_open_weight = 0.10000000149011612
    velocity_weight = 1.9782673120498657

    # Temperature parameters for transformations
    distance_temp = 3.867816686630249
    open_temp = 0.10000000149011612
    velocity_temp = 4.029726982116699

    # Compute the distance from the hand to the drawer
    hand_to_drawer_distance = torch.norm(drawer_grasp_pos -
        franka_grasp_pos, dim=-1)

    # Reward for minimizing the distance to the drawer
    distance_reward = torch.exp(-distance_temp * hand_to_drawer_distance
        ) * pos_distance_weight

    # Reward for opening the door, with normalized scaling
    open_reward = torch.exp(open_temp * torch.abs(cabinet_dof_pos[:, (3)] -
        cabinet_dof_pos[:, (3)].clamp(max=1.0))) * drawer_open_weight

    # Reward for positive velocity movements towards opening the drawer
    velocity_reward = torch.exp(velocity_temp * cabinet_dof_vel[:, (3)].
        clamp(min=0.0)) * velocity_weight

    # Total reward is a combination of the individual rewards
    total_reward = (2.102652072906494 * distance_reward + 1.0 * open_reward +
        1.9782673120498657 * velocity_reward)

    # Individual reward components stored in a dictionary for analysis
    reward_dict = {'distance_reward': distance_reward, 'open_reward':
        open_reward, 'velocity_reward': velocity_reward}

    return total_reward, reward_dict
```

**The Reward Function For Shadow-Hand-Kettle**

```python
from typing import Tuple, Dict
import math
import torch
from torch import Tensor
@torch.jit.script
def compute_reward(right_hand_pos: torch.Tensor, left_hand_pos: torch.
    Tensor, kettle_handle_pos: torch.Tensor, bucket_handle_pos: torch.
    Tensor, kettle_spout_pos: torch.Tensor, bucket_handle_rot: torch.Tensor,
    object_pos: torch.Tensor, object_linvel: torch.Tensor, object_angvel:
    torch.Tensor, goal_pos: torch.Tensor, goal_rot: torch.Tensor
    ) ->Tuple[torch.Tensor, Dict[str, torch.Tensor]]:
    distance_kettle_grasp = torch.norm(right_hand_pos - kettle_handle_pos,
        dim=-1)
    grasp_kettle_reward = 1.0 / (1.0 + distance_kettle_grasp)
    grasp_kettle_reward = 1.277540922164917 / (0.6857131719589233 + torch.
        norm(right_hand_pos - kettle_handle_pos, dim=-1))
    grasp_kettle_reward *= 0.9725923538208008
    grasp_bucket_reward = 1.3005189895629883 / (0.6645537614822388 + torch.
        norm(left_hand_pos - bucket_handle_pos, dim=-1))
    grasp_bucket_reward *= 0.9953349232673645
    alignment_tolerance = 0.34917423129081726
    spout_to_bucket_distance = torch.norm(kettle_spout_pos - goal_pos, dim=-1)
    spout_alignment_reward = torch.exp(-alignment_tolerance *
        spout_to_bucket_distance)
    pouring_velocity = torch.clip(object_linvel[..., 2],
        0.0010000000474974513, None)
    pouring_velocity_reward = torch.tanh(pouring_velocity) * 2.2268362045288086
    total_reward = (1.277540922164917 * grasp_kettle_reward +
        1.3005188703536987 * grasp_bucket_reward + 0.8290095329284668 *
        spout_alignment_reward + 1.221653938293457 *
        pouring_velocity_reward + 1.277540922164917 * grasp_kettle_reward)
    reward_components = {'grasp_kettle_reward': grasp_kettle_reward,
        'grasp_bucket_reward': grasp_bucket_reward,
        'spout_alignment_reward': spout_alignment_reward,
        'pouring_velocity_reward': pouring_velocity_reward,
        'grasp_kettle_reward': grasp_kettle_reward}
    return total_reward, reward_components
```

**The Critic Function For Franka-Cabinet**

```python
def calculate_distance(vec_a, vec_b):
    return ((vec_a - vec_b) ** 2).sum(axis=1).sqrt()
def get_correct_sub_traj_and_labels(trajectory_a, trajectory_b):
    traj_a_length = len(trajectory_a)
    traj_b_length = len(trajectory_b)
    assert traj_a_length == traj_b_length
    label_list = []
    for state_index in range(traj_a_length):
        state_a_info = trajectory_a[state_index]
        state_b_info = trajectory_b[state_index]
        # Calculate the manipulation distances
        lfinger_dist_a = calculate_distance(state_a_info['franka_lfinger_pos'],
            state_a_info['drawer_grasp_pos'])
        lfinger_dist_b = calculate_distance(state_b_info['franka_lfinger_pos'],
            state_b_info['drawer_grasp_pos'])
        rfinger_dist_a = calculate_distance(state_a_info['franka_rfinger_pos'],
            state_a_info['drawer_grasp_pos'])
        rfinger_dist_b = calculate_distance(state_b_info['franka_rfinger_pos'],
            state_b_info['drawer_grasp_pos'])
        # Cabinet DOF opening comparison (critical for success)
        cabinet_dof_a = state_a_info['cabinet_dof_pos'][:, 3]
        cabinet_dof_b = state_b_info['cabinet_dof_pos'][:, 3]
        # Check which trajectory has better performance at this index
        if all(lfinger_dist_a <= lfinger_dist_b) and all(rfinger_dist_a <=
            rfinger_dist_b) and all(cabinet_dof_a >= cabinet_dof_b):
            label_list.append(1)  # A is better
        elif all(lfinger_dist_a >= lfinger_dist_b) and all(rfinger_dist_a >=
            rfinger_dist_b) and all(cabinet_dof_a <= cabinet_dof_b):
            label_list.append(-1) # B is better
        else:
            label_list.append(0)

    return label_list
```

**The Critic Function For Shadow-Hand-Kettle**

```python
def get_correct_sub_traj_and_labels(trajectory_a, trajectory_b):
    import torch
    traj_a_length = len(trajectory_a)
    traj_b_length = len(trajectory_b)
    assert traj_a_length == traj_b_length
    label_list = []
    for state_index in range(traj_a_length):
        state_a = trajectory_a[state_index]
        state_b = trajectory_b[state_index]

        # Calculate distances for both trajectories
        right_hand_fingers_a = ['right_hand_ff_pos', 'right_hand_mf_pos', '
            right_hand_rf_pos', 'right_hand_lf_pos', 'right_hand_th_pos']
        right_hand_fingers_b = ['right_hand_ff_pos', 'right_hand_mf_pos', '
            right_hand_rf_pos', 'right_hand_lf_pos', 'right_hand_th_pos']
        left_hand_fingers_a = ['left_hand_ff_pos', 'left_hand_mf_pos', '
            left_hand_rf_pos', 'left_hand_lf_pos', 'left_hand_th_pos']
        left_hand_fingers_b = ['left_hand_ff_pos', 'left_hand_mf_pos', '
            left_hand_rf_pos', 'left_hand_lf_pos', 'left_hand_th_pos']

        # Calculate the right hand's distances to the kettle handle
        right_hand_distances_a = [torch.norm(torch.tensor(state_a[finger]) - torch.
            tensor(state_a['kettle_handle_pos']), p=2) for finger in
            right_hand_fingers_a]
        right_hand_distances_b = [torch.norm(torch.tensor(state_b[finger]) - torch.
            tensor(state_b['kettle_handle_pos']), p=2) for finger in
            right_hand_fingers_b]
        # Calculate the left hand's distances to the bucket handle
        left_hand_distances_a = [torch.norm(torch.tensor(state_a[finger]) - torch.
            tensor(state_a['bucket_handle_pos']), p=2) for finger in
            left_hand_fingers_a]
        left_hand_distances_b = [torch.norm(torch.tensor(state_b[finger]) - torch.
            tensor(state_b['bucket_handle_pos']), p=2) for finger in
            left_hand_fingers_b]
        # Calculate the distance between bucket handle and kettle spout
        spout_bucket_distance_a = torch.norm(torch.tensor(state_a['bucket_handle_pos
            ']) - torch.tensor(state_a['kettle_spout_pos']), p=2)
        spout_bucket_distance_b = torch.norm(torch.tensor(state_b['bucket_handle_pos
            ']) - torch.tensor(state_b['kettle_spout_pos']), p=2)
        # Determine the better trajectory at this state
        if all(ra <= rb for ra, rb in zip(right_hand_distances_a,
            right_hand_distances_b)) and \
            all(la <= lb for la, lb in zip(left_hand_distances_a,
                left_hand_distances_b)) and \
            spout_bucket_distance_a <= spout_bucket_distance_b:
             label_list.append(1)  # A is better
        elif all(ra >= rb for ra, rb in zip(right_hand_distances_a,
            right_hand_distances_b)) and \
             all(la >= lb for la, lb in zip(left_hand_distances_a,
                 left_hand_distances_b)) and \
             spout_bucket_distance_a >= spout_bucket_distance_b:
             label_list.append(-1)  # B is better
        else:
             label_list.append(0)  # Neither is better

    return label_list
```

