# OpenReview forum: "R*: Efficient Reward Design via Reward Structure Evolution and Parameter Alignment Optimization with Large Language Models"
_ICML.cc/2025/Conference — ICML 2025 poster_

### Official Review · Reviewer_apPp · 2025-03-10

**Overall Recommendation:** 3

**Summary:**

The paper introduces R*, an efficient framework for automatic reward function generation in reinforcement learning. R* addresses the challenge of designing high-quality reward functions by leveraging LLMs through two key components: reward structure evolution and parameter alignment optimization. The framework uses LLMs to generate modular reward functions, refines them with module-level crossover, and optimizes parameters via a critic-based voting mechanism for step-wise trajectory labeling. Experiments across eight robotic control tasks show that R* significantly outperforms existing methods, including Eureka and human-designed rewards, in both final policy performance and convergence speed.

**Claims And Evidence:**

yes

**Essential References Not Discussed:**

no

**Experimental Designs Or Analyses:**

This work conducts experiments in Isaac Gym and the Bidextrous Manipulation (Dexterity) benchmark, which are commonly used benchmarks previously.

**Methods And Evaluation Criteria:**

yes

**Other Comments Or Suggestions:**

typos: In line 6 of the pseudocode, are $F_{p1}$ and $F_{p2}$ sampled from $D_T$ or $P_{reward}$?

**Other Strengths And Weaknesses:**

Strengths
--

1. The paper is well-structured and the idea of parameter alignment optimization makes sence.
2. The implementation details and prompt designs are clearly and thoroughly described.
3. The experimental results are impressive.

Weaknesses
--

1. The proposed method heavily relies on several assumptions about the environments, including the need for environment code for LLM-based coding and detailed state information for state ranking. These requirements make it difficult to extend the method to more complex and real-world scenarios, such as image-based tasks. I understand that Eureka, the foundation of this method, has also been criticized for these limitations. I am curious whether the authors have considered strategies to address this issue, or at the very least, whether these limitations should be explicitly discussed in the manuscript.
2. The reported Eureka performance seems significantly lower than that reported in the original Eureka paper.

**Questions For Authors:**

1. I am uncertain about the crossover operation: How are the parents selected, and how is it determined which reward module to insert into another parent?
2. Which reward functions are used for training in Figure 2?
3. Are the trajectories for parameter alignment optimization sampled using the initially randomized policy?
4. Are the generated reward functions only suitable for PPO?

**Relation To Broader Scientific Literature:**

This work builds on existing LLM-based reward design methods and introduces technical enhancements to improve their performance.

**Theoretical Claims:**

There is not theoretical claim in this submission.

---

> ### Author Rebuttal · Authors · 2025-04-01
>
> We thank the reviewer for the constructive feedback. For the reviewer's questions, we will respond to them one by one as follows.
>
> 1. **[How are the parents selected, and how is it determined which reward module to insert into another parent?]**
>
> We maintain a buffer of reward functions with a fixed size of 5 (Only the top-5 best-performing ones are retained), where each reward function is associated with the success rate of the RL policy it guides. We apply a softmax over these success rates and sample two reward functions based on the resulting probabilities.
>
> We then randomly select reward modules for crossover. The main advantage of this approach is that it does not require any API calls. A potential improvement would be to leverage an LLM to guide the module selection process; however, this typically incurs additional LLM API usage.
>
> 2. **[Which reward functions are used for training in Figure 2?]**
>
> The reward function that achieves the highest RL success rate across all generations is selected for final evaluation, which is consistent with the setting used in Eureka.
>
> 3. **[Are the trajectories for parameter alignment optimization sampled using the initially randomized policy?]**
>
> The trajectories are collected using the trained policies guided by different reward functions within the population.
>
> 4. **[Are the generated reward functions only suitable for PPO?]**
>
> The generated reward functions are applicable to various RL algorithms. To demonstrate this, we conducted experiments on Metaworld, using the generated rewards to guide SAC for policy learning (max success rate and the required steps). The results are as follows:
>
> |  | **drawer-open** | **button-press** | **window-close** |
> | --- | --- | --- | --- |
> | Expert Reward | 100% \| 33600 | 100% \| 399000 | 100% \| 220500 |
> | R* | 100%  \| 36720 | 100%  \| 421200 | 100% \| 772050 |
>
> We observe that **R*** achieves a 100% success rate on these tasks, matching the performance of the expert-designed rewards.
>
> 5. **[I am curious whether the authors have considered strategies to address this issue, or at the very least, whether these limitations should be explicitly discussed in the manuscript.]**
>
> For complex visual tasks, we believe a feasible direction is to obtain the relative positions of targets through object detection and prediction, and associate them with variables—this typically requires additional training. However, our current method does share the limitation mentioned by the reviewer: it cannot be applied to tasks where low-level information is inaccessible.
>
> 6. **[The reported Eureka performance seems significantly lower than that reported in the original Eureka paper.]**
>
> The main reason lies in our environment configuration, which limits the number of parallel environments. In the original Eureka paper, some tasks use a large number of parallel environments, requiring at least 4 RTX 4090 GPUs to run. We reduce the number of parallel environments to ensure that the program can run with only 40GB of GPU memory. However, this reduction significantly increases the learning difficulty for the algorithm.
> Besides, we use GPT-4o for our experiments, which may also introduce some differences.
>
> However, to ensure a fair comparison, all experiments are conducted under the same configuration.
>
> The specific configuration of the number of parallel environments is shown in the table below.
>
> |  | Franka-Cabinet | Swing-Cup | Hand-Over | Hand-Scissors | Allegro-Hand | Door-Open-Outward | Kettle | Pen |
> | --- | --- | --- | --- | --- | --- | --- | --- | --- |
> | Env number | 4096 | 256 | 512 | 128 | 1024 | 2048 | 128 | 256 |
>
> 7. **[typos: In line 6 of the pseudocode]**
>
> Thank you to the reviewer for pointing out this typo. We will correct it in the revised revision.
>
> ---
>
> **We would appreciate it if the reviewer can confirm that the concerns had been addressed and, if so, reconsider the assessment. We’d be happy to engage in further discussions.**

---

> > ### Comment · Reviewer_apPp · 2025-04-04
> >
> > Thank you to the authors for their response. While some of my concerns have been addressed, I still have a few remaining questions:
> > - What is the experimental setup for the MetaWorld experiments with SAC? Specifically, is SAC used as the training algorithm during iterative evolution?
> > - I understand the discrepancy between the reported Eureka performance and that in the original Eureka paper. However, my concern about the comparability of the results remains, as the difference is quite substantial. I believe it is necessary to reproduce both Eureka and the proposed method under a similar setup used in Eureka. This is also important for evaluating the scalability of the proposed method across different LLMs and computational resources.
> > - As acknowledged by the authors, the proposed method cannot be applied to tasks where low-level information is inaccessible. I consider this a significant limitation that restricts the method’s applicability, even though this constraint is shared by many prior works. I hope the authors will provide a serious discussion of this issue in the paper, along with a convincing realistic justification for the current experimental setup.
> >
> > If these concerns are addressed, I would be happy to raise my score to acceptance.
> >
> > **Update: Thank you for the authors' response. I have updated my score from 2 to 3. I believe the newly added experiments and discussions have improved the quality of the paper.**

---

> > > ### Author Response · Authors · 2025-04-09
> > >
> > > 1. **[What is the experimental setup for the MetaWorld experiments with SAC? Specifically, is SAC used as the training algorithm during iterative evolution?]**
> > >
> > > We replace the PPO with SAC. the population size is set to 5. Each evaluation is performed after 200,000 environment steps. Our experiments primarily demonstrate that our method are also capable of generating rewards to effectively guide the learning and optimization of other RL algorithms, achieving performance that is competitive with expert-designed rewards.
> > >
> > > 2. **[Reproduce both Eureka and the proposed method under a similar setup used in Eureka]**
> > >
> > > Thank you for the reviewer’s suggestion. One of the key factors in performing comparisons under the original setting is hardware resources. Due to limitations of our existing servers, we were unable to successfully run most tasks, which typically require over 60 GB of GPU memory, and in some cases up to 100 GB.
> > >
> > > To address the reviewer’s concern, we rent additional servers to carry out the experiments. **We strictly follow the original settings**, using GPT-4-0314 as the LLM model. The avg success rates are as follows:
> > >
> > > |  | **Franka**  | **Swing-Cup** | **Hand-Over** | **Kettle** | **Scissor** | **Door-Open -outward** |
> > > | --- | --- | --- | --- | --- | --- | --- |
> > > | Eureka | 33% | 53% | 83% | 70% | 100% | 98%   |
> > > | R* | 73% | 96% | 93% | 95% | 100%  | 100%  |
> > >
> > > From the results, we observe that R* also outperforms Eureka under the original settings.
> > >
> > > 3. **[A serious discussion of this issue in the paper, along with a convincing realistic justification for the current experimental setup. ]**
> > >
> > > Thank you for the reviewer’s valuable suggestion. Our experiments primarily focus on manipulation and dexterous hand control tasks. In real-world robotic applications, it is typically feasible to directly access various low-level states from the robot itself. As for the information regarding external objects, it can usually be obtained using sensors such as LiDAR and depth cameras.
> > >
> > > For robot control, a real2sim2real paradigm is commonly adopted. Since real-world information from the robot and its environment are available, training can proceed by aligning the information from the real robot with that in simulation. This enables direct deployment of the trained policies in real-world scenarios.
> > >
> > > For tasks where low-level information is inaccessible—such as non-invasive control tasks involving software operation or game play—only visual observations (i.e., images) are typically available. We believe that the key to applying R* in such cases lies in effective information extraction. In these scenarios, A detection model (e.g., YOLO) can be used to identify and extract relevant features from visual input. Once these key features are obtained, the subsequent reward generation and policy training processes are consistent with those used in tasks where low-level information is accessible.
> > >
> > > We appreciate the reviewer’s insightful comment and will include a detailed discussion of this issue in the revised version. We welcome any further suggestions from the reviewer and are happy to incorporate further discussion as needed.
> > >
> > > ---
> > >
> > > We hope that the above response and experiments can address the concern raised by the reviewer.
> > >
> > > We sincerely appreciate the valuable time and suggestions provided by the reviewer throughout the entire review process.
> > >
> > > ---
> > >
> > > **Author Response:**
> > >
> > > **We are delighted to have addressed the reviewer’s concerns and sincerely appreciate the recognition and support for our work!**
> > >
> > > **Thank you again for your constructive comments and the in-depth discussions, which have helped us strengthen our work.**

---

### Official Review · Reviewer_268i · 2025-03-13

**Overall Recommendation:** 3

**Summary:**

This paper proposes a new method for designing reward functions with LLMs, R*. R* uses LLMs to generate modular reward function components, and maintains a population of reward functions. These population of reward functions are evaluated based on how well they guide the agent to the sparse reward. Based on their fitness, these reward functions undergo mutation using the LLM. Furthermore, they do parameter alignment of the parameters in the reward function using a pairwise preference loss.

**Claims And Evidence:**

The authors claims about prior works in the introduction were not fully supported. For example, in line 53 the authors say that the works they build off face instability and inaccuracies in parameter configuration. However, this is not explained in depth and it seems to be a crucial part of their motivation.

The authors claims about superior performance compared to eureka are well supported by many experiments across different settings.

**Essential References Not Discussed:**

This paper does not really discuss in depth their differences with Eureka (Ma 2024). I think this is necessary, as they seem to directly build upon this work. This makes the contribution of this paper difficult to ascertain.

**Experimental Designs Or Analyses:**

The experimental design seems quite sound to me. In all experiments they compare with SOTA baselines (eureka). They also conduct experiments over 5 random seeds and report the mean and standard deviation in their learning curves. Finally, they keep the hyperparameter setting to be very similar to Eureka, which makes it likely to be a fair comparison.

**Methods And Evaluation Criteria:**

Yes the proposed methods and evaluation criteria make a lot of sense. Their work tries automatically design reward functions, and conducts experiments in many popular RL environments to evaluate their approach. Their evaluation based on success rate makes sense as well. In general they try to improve upon Eureka, and use a very similar setting to that work.

**Other Comments Or Suggestions:**

Typo in line 337.

**Other Strengths And Weaknesses:**

Strengths:
- The methodology is seemingly novel, although that is hard to judge.
- The experimental results are comprehensive, and their method shows clear improvements.

Weaknesses:
- The writing is not that well structured. For example paragraphs 2 and 3 are very long and hard to read.
- This paper mainly seems like a follow up work on Eureka. However, the authors do not clearly state their contribution compared to eureka, or state the benefits of their method. They only use vague language such as prior works have “instability and inaccuracy” (line 54) but do not say concretely how they improve upon it.

**Questions For Authors:**

Can you provide a more in depth comparison to Eureka?

**Relation To Broader Scientific Literature:**

This paper does not do a good job discussing their place in the broader literature. To me it seems like they make novel improvements to Eureka, which is a popular framework. However, they do not discuss Eureka in depth, so it is really hard to estimate the novelty and contribution of this work.

**Theoretical Claims:**

na

---

> ### Author Rebuttal · Authors · 2025-04-01
>
> We thank the reviewer for the constructive feedback and for a positive assessment of our work. For the reviewer's questions, we will respond to them one by one as follows.
>
> 1. **[they do not discuss Eureka in depth, so it is really hard to estimate the novelty and contribution of this work. ]**
>
> Eureka maintains a population of reward functions and iteratively improves them using an LLM. The improvement process involves feeding the LLM with the best-performing reward function discovered in the current population, along with feedback from its RL training process. Based on this information, the LLM adjusts both the reward function logic and its parameters to enhance performance.
>
> The limitations of Eureka are mainly reflected in two aspects:
>
> 1. **Insufficient utilization of existing knowledge**: Eureka relies on the best-performing individual during each reward function update. However, the individual may represent suboptimal solution, and continuously optimizing based on it can lead to poor results.
> 2. **Inefficient parameter optimization**: Reward functions often involve numerous parameters, such as weights within and between reward components. Iterative optimization of these parameters using LLM is inefficient.
>
> To address the two issues mentioned above, we propose **reward structure evolution** and **parameter alignment optimization**. The former fully leverages high-quality reward functions without introducing any API calls, enabling thorough exploration through crossover among components within superior reward functions. The latter adjusts parameters via preference learning, where a key challenge lies in constructing a reliable preference dataset. To tackle this, we utilize the LLM to generate critic functions—a process that does not involve human participation. Moreover, to ensure the accuracy of preference labels, we build a population of critics and annotate preferences between states through a voting mechanism. Finally, we optimize the reward function parameters based on the learned preferences, providing more accurate reward signals.
>
> 1. **[The writing is not that well structured. For example paragraphs 2 and 3 are very long and hard to read.]**
>
> Thank you for the reviewer’s suggestion. We will improve the presentation in the revised version.
>
> ---
>
> **We would appreciate it if the reviewer can confirm that the concerns had been addressed and, if so, reconsider the assessment. We’d be happy to engage in further discussions.**

---

### Official Review · Reviewer_38wQ · 2025-03-13

**Overall Recommendation:** 3

**Summary:**

This paper introduces R* which designs reward function by utilizing LLMs to construct a set of reward functions, 'evaluating' these rewards by training PPO agents to maximize the rewards, and then improving the reward functions based on voting mechanism followed by preference-based learning. Ablation study shows that the proposed idea of using crossover operator (for better exploration of reward design space) and parameter alignment optimization (based on voting and preference learning) are indeed effective. Experiments are conducted on IsaacGym and Dexterity benchmarks and the main baseline is Eureka that is based on the same high-level idea of generating rewards with LLM, evaluating rewards, and improving rewards based on reflection.

## update after rebuttal

My score has been updated from 2 to 3 during the rebuttal process. My acceptance recommendation is conditional on the authors' promise to largely update the manuscript (especially Introduction) for improving the positioning of the paper.

**Claims And Evidence:**

In terms of evaluating the efficacy of the newly proposed techniques, the paper provide an ablation study that supports it. But this paper is not making that much of a claim with regard to 'hypothesis that explains why this method should work better than previous algorithm'.

**Essential References Not Discussed:**

N/A

**Experimental Designs Or Analyses:**

The performance of PPO with oracle reward seems very low to me -- it's much lower than the performance reported in Eureka tasks. Maybe I'm missing something here but I'm not sure why this happens.

**Methods And Evaluation Criteria:**

Eureka also used IsaacGym tasks so it makes sense. Some qualitative/quantiative analysis on the quality of learned reward is missing.

**Other Comments Or Suggestions:**

N/A

**Other Strengths And Weaknesses:**

One weakness of this paper is that writing is very verbose and some paragraphs are really long. In particular the ones in the introduction.

**Questions For Authors:**

- The biggest weakness of this paper is that it's not properly positioning the contributions of this paper with regard to existing works, and just say that "reward design is challenging and we did a bunch of things to do that!". But the very high-level idea of this work is very similar to Eureka's. Using LLMs to generate reward functions, training policies, and improvement based on reflection. So what's the fundamental limitation of Eureka and how is the proposed idea addressing the limitations? Why should the proposed algorithm work better than Eureka?
- Would it be possible to provide quantitative/qualitative analysis that compares the quality of learned rewards, instead of only showing the downstream performance?
- Why is performance with Oracle reward function worse than the ones reported in Eureka paper?

**Relation To Broader Scientific Literature:**

Key contributions of this paper will be very technical contributions around improving many parts of Eureka. But motivations for introducing these components are very weak so it is not clear if this is of interest to broader literature.

**Theoretical Claims:**

N/A

---

> ### Author Rebuttal · Authors · 2025-04-01
>
> We thank the reviewer for the constructive feedback. For the reviewer's questions, we will respond to them one by one as follows.
>
> ---
>
> 1. **[The performance of PPO with oracle reward seems very low to me -- it's much lower than the performance reported in Eureka tasks. Maybe I'm missing something here but I'm not sure why this happens.]**
>
> The main reason lies in our environment configuration, which limits the number of parallel environments. In the original Eureka paper, some tasks use a large number of parallel environments, requiring at least 4 RTX 4090 GPUs to run. We reduce the number of parallel environments to ensure that the program can run with only 40GB of GPU memory. However, this reduction significantly increases the learning difficulty for the algorithm.
>
> However, to ensure a fair comparison, all experiments are conducted under the same configuration.
>
> The specific configuration of the number of parallel environments is shown in the table below.
>
> |  | Franka-Cabinet | Swing-Cup | Hand-Over | Hand-Scissors | Allegro-Hand | Door-Open-Outward | Kettle | Pen |
> | --- | --- | --- | --- | --- | --- | --- | --- | --- |
> | Env number | 4096 | 256 | 512 | 128 | 1024 | 2048 | 128 | 256 |
>
> 2. **[One weakness of this paper is that writing is very verbose and some paragraphs are really long. In particular the ones in the introduction.]**
>
> Thank you for the valuable suggestions. We will work on improving the overall clarity and expression throughout the manuscript in the revised revision.
>
> 3. **[What's the fundamental limitation of Eureka and how is the proposed idea addressing the limitations? Why should the proposed algorithm work better than Eureka?]**
>
> The limitations of Eureka are mainly reflected in two aspects:
>
> 1. **Insufficient utilization of existing knowledge**: Eureka relies on the best-performing individual during each reward function update. However, the individual may represent suboptimal solution, and continuously optimizing based on it can lead to poor results. The introduction of a crossover mechanism aims to fully leverage existing knowledge without introducing any API calls, enabling integrated exploration across different modules of the superior reward functions and ensuring more efficient exploitation.
> 2. **Inefficient parameter optimization**: Reward functions often involve numerous parameters, such as weights within and between reward components. Iterative optimization of these parameters using LLM is inefficient. To address this, we propose a preference learning-based approach, which makes the gradients of the reward function parameters differentiable. By collecting preference data from the critic population, we optimize these parameters more effectively.
>
> To solve above problems, we propose the **reward structure evolution and parameter alignment Optimization**. The former focuses on efficient structure search to avoid getting trapped in suboptimal designs, while the latter performs effective parameter optimization. By combining both components, our method enables more efficient reward design compared to Eureka.
>
> ---
>
> We would appreciate it if the reviewer can confirm that the concerns had been addressed and, if so, reconsider the assessment. We’d be happy to engage in further discussions.

---

> > ### Comment · Reviewer_38wQ · 2025-04-03
> >
> > I have updated the score 2 to 3. Please make sure to re-write introduction to properly discuss prior works and the main contributions of the paper, and clarify the different experimental setup.

---

> > > ### Author Response · Authors · 2025-04-03
> > >
> > > We are delighted to have addressed the reviewer’s concerns and sincerely appreciate the recognition and support for our work!
> > >
> > > As requested, we will provide a comprehensive discussion of the related work and main contributions in the revised manuscript, and offer a clearer explanation of the experimental setup.
> > >
> > > Thank you again for your constructive comments and the in-depth discussions, which have helped us strengthen our work.

---

### Official Review · Reviewer_CcRx · 2025-03-16

**Overall Recommendation:** 3

**Summary:**

This paper proposes LLM-based reward function generation to train models for tasks such as robotic hand manipulation.

The presented method can be decomposed into:
1. generation of modular reward functions using an LLM
2. augmentation with new reward functions based on modular mixing of the functions from step 1
3. Generation of trajectories and ratings from an LLM-generated critic population
4. Reward function parameter optimization based on the labeled trajectories
5. PPO-based training of agents for the end task
6. LLM-based reflection based refinement of the reward functions.

**Claims And Evidence:**

Claims made: the proposed method outperforms existing SOTA methods for reward function design

**Essential References Not Discussed:**

N/A

**Experimental Designs Or Analyses:**

The experimental design follows standard methodology and is very similar to existing methods that address reward function design.

**Methods And Evaluation Criteria:**

Yes, the proposed methods and evaluation criteria are standard for the premise of the problem considered.

**Other Comments Or Suggestions:**

N/A

**Other Strengths And Weaknesses:**

Strengths:

- The paper's results are very strong and are significantly better than the existing SOTA methods.

Weaknesses:

- The novelty of the proposed ideas is low, and seem to be a collection of multiple ad-hoc steps
- The writing can be improved. For example, the paper is focused mainly of robotic hand manipulation tasks, and there is no mention of this until the experiments section. The paper should do a better job of providing context and motivation.

**Questions For Authors:**

1. It seems to me that the critic population annotation is more important to the success of the method than crossover-based reward function generation. Is this correct?

2. What is the intuition behind the crossover-based design? I am unable to get a feel for why it leads to such a big improvement in performance, and why the LLMs cannot generate such functions in the first place.

**Relation To Broader Scientific Literature:**

The paper improved upon previous SOTA methods (https://arxiv.org/pdf/2310.12931) that also uses LLMs for designing reward functions.

**Theoretical Claims:**

N/A

---

> ### Author Rebuttal · Authors · 2025-04-01
>
> We thank the reviewer for the constructive feedback and for a positive assessment of our work. For the reviewer's questions, we will respond to them one by one as follows.
>
> 1. **[The novelty of the proposed ideas is low, and seem to be a collection of multiple ad-hoc steps]**
>
> Our work focuses on reward function construction and enhancement from two key perspectives: **reward structure evolution** and **parameter alignment optimization**. The former aims to design meaningful reward components using LLMs and evolutionary principles, while the latter focuses on optimizing the parameters within those reward components.
>
> In **reward structure evolution**, we introduce code-level crossover operations to fully exploit existing reward function implementations and address potential suboptimality issues.
>
> In **parameter alignment optimization**, we first use LLMs to construct state-level critic functions. To mitigate potential errors from relying on a single critic, we build a critic population and use it to generate a preference dataset. We then make the numerical values in the reward function code differentiable and apply preference learning to efficiently optimize the reward function.
>
> Our method is primarily designed and optimized based on the challenges associated with reward generation, aiming to construct more efficient reward functions.
>
> 2. **[the paper is focused mainly of robotic hand manipulation tasks, and there is no mention of this until the experiments section. The paper should do a better job of providing context and motivation.]**
>
> Thank you for the reviewer’s suggestion. We will improve the presentation in the revised version.
>
> 3. **[It seems to me that the critic population annotation is more important to the success of the method than crossover-based reward function generation. Is this correct?]**
>
> The goal of crossover-based reward function generation is to ensure the effective utilization of existing reward functions, which can  facilitate the discovery of well-structured designs. Since parameter optimization does not modify the underlying structure, it is generally most efficient to apply parameter tuning only after a reasonable structure has been identified.
>
> In many tasks, the initial parameter settings provided by the LLM are already fairly reasonable. In such cases, parameter alignment may have limited impact, whereas crossover can continuously explore and recombine high-quality existing reward functions to uncover improved designs. However, when the LLM provides suboptimal parameter settings, parameter optimization becomes crucial for constructing more effective reward guidance.
>
> Based on our experimental results, parameter alignment is generally more efficient in most cases.
>
> 4. **[What is the intuition behind the crossover-based design? I am unable to get a feel for why it leads to such a big improvement in performance, and why the LLMs cannot generate such functions in the first place.]**
>
> Eureka continuously improves the reward function by providing the best-performing individual to the LLM, while other reward functions are directly discarded. When the best individual is actually suboptimal, reflecting and improving based on it can lead the entire population to converge toward a suboptimal solution, ultimately resulting in low-quality outcomes.
>
> In contrast, the crossover-based design aims to fully leverage existing superior reward functions without introducing any additional LLM overhead. By performing crossovers between different reward modules in superior reward functions, it enables thorough exploration and helps avoid being trapped in suboptimal solutions.
>
> As shown in the results of Figure 4, the reward functions discovered by EA have a significantly higher probability of being the best in the population (the probability of best policies originating from crossover exceeds 50% in most tasks,with some
> tasks surpassing 80%), which further demonstrates EA's ability to effectively explore existing rewards and discover better reward functions.
>
> ---
>
> **We would appreciate it if the reviewer can confirm that the concerns had been addressed and, if so, reconsider the assessment. We’d be happy to engage in further discussions.**

---

### Decision · Program_Chairs · 2025-05-01

**Decision:**

Accept (poster)

**Comment:**

The submission proposes an effective automated reward design framework, R*, which improves upon existing methods, in reward structure evolution and parameter alignment optimization. Reviewers praised the strong empirical results and comprehensive experiments conducted on robotic control tasks. Initial concerns regarding clarity, novelty, and experimental setups were thoroughly addressed during rebuttal. The authors provided additional experiments and improved manuscript structure. Given these improvements and the paper's clear contributions, the AC agrees with the reviewers to accept the submission.